# Jointly Training Large Autoregressive Multimodal Models

**Emanuele Aiello**[†,◇,*]**Lili Yu**[◇] **, Yixin Nie**[◇] **, Armen Aghajanyan**[◇] **, Barlas Oguz**[◇]
[†]Politecnico di Torino, [◇]Meta AI

## Abstract

In recent years, advances in the large-scale pretraining of language and text-to-image models have revolutionized the field of machine learning. Yet, integrating these two modalities into a single, robust model capable of generating seamless multimodal outputs remains a significant challenge. To address this gap, we present the Joint Autoregressive Mixture (JAM) framework, a modular approach that systematically fuses existing text and image generation models. We also introduce a specialized, data-efficient instruction-tuning strategy, tailored for mixed-modal generation tasks. Our final instruct-tuned model demonstrates unparalleled performance in generating high-quality multimodal outputs and represents the first model explicitly designed for this purpose.

## 1 Introduction

Autoregressive text-to-image models, as exemplified by works such as Yu et al. (2023; 2022), have made remarkable strides in generating highly detailed images, paralleling the achievements of Diffusion Models Nichol et al. (2022); Ramesh et al. (2022); Rombach et al. (2022). These models bear architectural resemblance to Large Language Models (LLMs), yet their training regimen is tailored for paired image-text data. LLMs on the other hand (Brown et al., 2020; Zhang et al., 2022; Touvron et al., 2023) are limited to text-based output, thus lacking multimodal generative capabilities despite their proficiency in textual tasks. The subfield of Multimodal Large Models has emerged in recent years Tsimpoukelli et al. (2021); Alayrac et al. (2022); Li et al. (2022a) in the quest to bring together the disparate strengths of vision and language models. Despite important advances in this direction, these models still predominantly generate one modality, thereby constraining their expressiveness. This study aspires to break this limitation by developing a multimodal model capable of generating integrated text and image outputs.

To achieve this objective, we conduct a comprehensive empirical investigation into the fusion of two specialized autoregressive, decoder-only, large transformer models, each designed for unique tasks (one for text-to-image and a text only model). We introduce a set of methods under the umbrella of the Joint Autoregressive Mixture (JAM) framework. In building this framework, we take advantage of the inherent architectural compatibility of autoregressive text-to-image models with LLMs, allowing us to do deep model fusion and joint training in ways which would otherwise not be possible. Our modular and data-efficient solution allows for deep, rapid and effective integration of continually evolving large models, using less than 1% of the original pretraining data for both parent models.

Our contributions to this study are twofold. First, we establish the feasibility of blending autoregressive text-to-image models with LLMs into a unified architecture that retains the core strengths of each while revealing new, emergent capabilities. Second, we present innovative strategies for multimodal instruction tuning, utilizing text-based instructions and a custom-curated dataset designed explicitly for image generation. The result is a first-of-its-kind large multimodal model which can coherently generate long-form content with interleaved text and images.

## 2 Methods

To tackle the challenge of creating a unified model that excels at vision-language generative tasks, we propose to combine two autoregressive decoder-only architectures. Our primary image-text model is

---

[*]work done as an intern in Meta AI, correspondence <emanuele.aiello@polito.it>

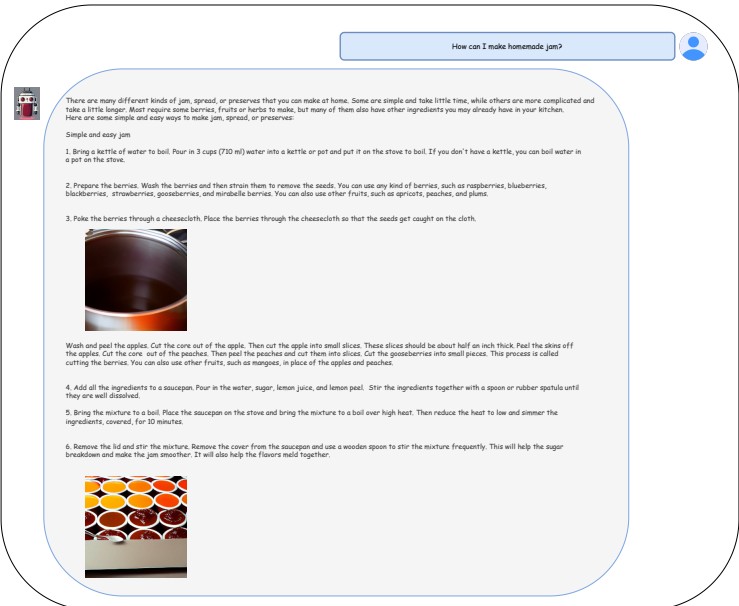

Figure 1: Selected sample generated by our instruction-tuned JAM-Cross model. The model can complex mixed-modal outputs with coherent alignment between generated text and images.

CM3leon (Yu et al., 2023), trained on 2.4T image-text caption tokens. In contrast, using the same architecture, our LLM (Molybog et al., 2023) has been trained on 1.4T text tokens. Both models have 7B parameters, we provide additional architectural details in Section 3.1. Our overall methodology develops in two stages. In the first stage (Sect. 2.1), we first combine and align the models. In the second stage (Sect. 2.2), we explore new directions for instruction tuning focused on interleaved image-text generation.

## 2.1 CONTINUED PRETRAINING

We combine the two pretrained models into a singular, cohesive structure in our proposed framework. This composite model is fine-tuned using a hybrid dataset comprising both text-only and image-text samples within our *continued pretraining* phase. The central motivation behind this approach is to seamlessly merge the capabilities of two pretrained models, capitalizing on the unique strengths of each. The training procedure is data-efficient since the original pretrained models are typically trained on trillions of tokens. In contrast, our procedure only uses 50 billion tokens, corresponding to 1.3% of the total data used during the pretraining.

### 2.1.1 MODEL MERGING

The concept of model merging has been previously utilized to combine models that share identical optimization trajectories (Kaddour et al., 2022), or models that are trained on identical datasets but have independent optimizations (for instance, Matena & Raffel (2022); Wortsman et al. (2022); Ainsworth et al. (2022)). A consistent approach across these studies is to combine models without any training. Our approach diverges from this convention; we view the merged model as a powerful initialization for subsequent training on mixed-modal data. The weights of the averaged model are defined as:

$$\boldsymbol{\theta}_{average} = \frac{1}{2}\boldsymbol{\theta}_{llm} + \frac{1}{2}\boldsymbol{\theta}_{img} \tag{1}$$

Where $\boldsymbol{\theta}_{llm}$ and $\boldsymbol{\theta}_{img}$ represent the weights of the LLM and the text-to-image model respectively. In this study, we explore weights merging specifically to multimodal decoder-only large transformer models, and notably, on an unprecedented scale, involving models trained on trillions of tokens from diverse datasets. In the following sections, we refer to our average model as JAM-Uniform.

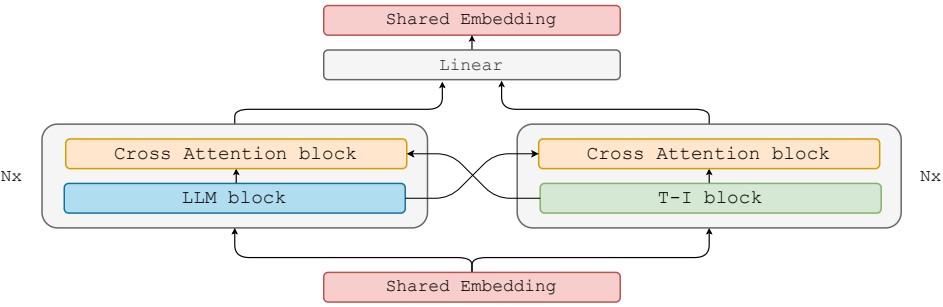

Figure 2: JAM-Cross, architecture overview. The cross-attention blocks are interleaved between the original LLM block and the Text-Image blocks, and the output embedding between the two branches are concatenated and then projected to the output embedding dimension.

### 2.1.2 WIDTH CONCATENATION

Our second approach employs the pretrained weights to initialize a wider architecture. Our new model has hidden dimensions $d_{joint} = 8192$, which is doubled with respect to one of the two original models $d_{llm} = d_{img} = 4096$. We keep the same number of layers of the original architectures. The resulting architecture has 26B parameters, initialized starting from the pretrained weights of our backbones. The token embedding input/output projections and the learned positional embeddings of the two initial models are concatenated on the hidden dimension. The attention weights (e.g query projection) $W_{q,combined} \in \mathbb{R}^{d_{joint} \times d_{joint}}$ are initialized as:

$$W_{q,combined} = \begin{pmatrix} W_{q,llm} & W_{q,llm} \\ W_{q,img} & W_{q,img} \end{pmatrix} \tag{2}$$

Where $W_{q,llm}, W_{q,img} \in \mathbb{R}^{d_{llm} \times d_{llm}}$ represent the weights for the query projection of a generic attention layer. All the other weights (FFNs and output projections) are initialized following the same logic. We also experiment with slight variations of the approach:

$$W_{q,combined} = \begin{pmatrix} W_{q,llm} & W_{q,average} \\ W_{q,img} & W_{q,average} \end{pmatrix} \tag{3}$$

Instead of copying the two models' parameters, we use the average to initialize half of the new parameters. We name the resulting model JAM-Width.

### 2.1.3 CROSS MODEL FUSION

We propose to embed cross-attention layers between the foundational models to facilitate seamless information interchange while preserving the original models' knowledge. Given two decoder-only transformers models $\mathcal{T}_{llm}$ and $\mathcal{T}_{img}$, we introduce a bi-directional cross-attention mechanism that enables the layers of one model to attend to the corresponding layer's output of the other model. This approach allows for a progressive exchange of information at different representation levels. For a specific layer $l$, let the models produce sequences of hidden states $H_{llm,l}$ for $\mathcal{T}_{llm}$ and $H_{img,l}$ for $\mathcal{T}_{img}$ where these hidden states are outputs from layer $l$. The output of the cross-attention mechanism ($H_{cross,l}$) from $\mathcal{T}_{img} \rightarrow \mathcal{T}_{llm}$ for a given layer is evaluated as:

$$Q_{cross,l} = W_{q,l} H_{llm,l-1}, \quad K_{cross,l} = W_{k,l} H_{img,l-1}, \quad V_{cross,l} = W_{v,l} H_{img,l-1} \tag{4}$$

$$H_{cross,l} = \text{Softmax}\left(\frac{Q_{cross,l} K_{cross,l}^T}{\sqrt{d_k}}\right) V_{cross,l} \tag{5}$$

Where $W_q, W_k, W_v$ represent the query, key, and value projection weights of the newly inserted cross-attention layers. A symmetric process is applied for the reverse direction $\mathcal{T}_{llm} \rightarrow \mathcal{T}_{img}$. We use a shared input-output projection layer, initializing the weights of the text tokens from the LLM input embedding and the weights of the image tokens from the image-text model. We insert a new linear projection layer that takes the concatenation of the two model's output embeddings as input.

Figure 2 illustrates a schematic of our model configuration. We refer to the model resulting from this approach as JAM-Cross. Additional architectural details and the underlying design choices can be found in Sect. 3.1. The ablation study for the optimal frequency of inserting new layers is presented in Sect. 3.3.

## 2.2 MULTIMODAL CONVERSATIONAL INSTRUCT TUNING

Supervised fine-tuning is a fundamental tool to leverage the abilities of large pretrained models. Recently, instruct tuning has been extended to a multimodal setting (Liu et al., 2023; Dai et al., 2023); however, all the existing approaches are focused on visual understanding abilities. In this work, we study instruction tuning tailored to interleaved image-text generation.

We collect a small and curated mixed-modal dataset to teach our JAM model to support textual explanations with coherent images. Since in the first stage, the model has been trained on image-text captions and text-only data; we train on interleaved image-text data during this phase. Our approach is inspired by the Superficial Alignment Hypothesis from LIMA (Zhou, 2023), which posits that a model's foundational knowledge and skills are entirely learnt during the pretraining. Instruction tuning is then used to guide the model in selecting the appropriate subdistribution of format used when interacting with users. Our results demonstrate that the model can quickly learn the style of images and text from a small curated dataset, suggesting that LIMA hypotesis holds not only for learning the text style but also for images. In our experiments, we consider two slightly different instruction tuning settings, we introduce a small portion of the image-text Shutterstock data with retrieval augmentation and we find this approach beneficial to preserve the generated image quality when generating with retrieval augmentation. Sect 3 presents a comparison between these two strategies. We train using a standard supervised procedure without leveraging any reinforcement learning or human preference strategy. In this instruction-tuning phase, we leverage interleaved image-text data in contrast to previous methods (Koh et al., 2023a) that rely only on image-text caption and no instruction tuning, our experimental results confirm the benefits of training with interleaved image-text data.

## 3 EXPERIMENTS

### 3.1 EXPERIMENTAL DETAILS

**Tokenizers** For images, we use the VQ-VAE tokenizer from Gafni et al. (2022). The image resolution is set to $256 \times 256$, $1024$ tokens represent each image, and the vocabulary has a size of $8192$. Our text tokenizer is the same that have been used to train the two parent models, trained over the Zhang et al. (2022) data for text. We introduce the additional <break> token used by CM3leon to identify a modality break.

**Image-Text Autoregressive Model** We adopt CM3leon as the image-text autoregressive backbone. The model has a standard decoder-only architecture with some peculiarities: no bias terms, dropout, and learnable parameters for layer norms. It has been trained on 2.4T image-text tokens and uses a sequence length 4096.

**LLM** As an LLM backbone, we select a model with the same architecture as CM3leon, trained in Molybog et al. (2023) this allows us to experiment with a broader range of approaches, such as weight averaging and width concatenation. The model is trained on 1.4T text data with a 2048 context length, and we further fine-tuned it with a 4096 context length using only 30B text tokens.

**Objective** In all our experiments, we employ the CM3 objective introduced in Aghajanyan et al. (2022); this objective accepts the original sequence as input or transforms it into an infilling instance by masking specific spans and relocating them to the end of the document. Then, the model is optimized for minimizing the standard autoregressive loss $-\log p(x_{input})$. This objective allows for optional bidirectionally and increases the versatility of the model that can be used for both infilling or standard autoregressive generation. We prevent the objective from masking across the modality <break> tokens.

**Retrieval Augmentation** We employ multimodal retrieval augmentation introduced in Yasunaga et al. (2022) for our training procedure. The retrieval procedure employs a dense retriever $r$, a memory bank $\mathcal{M}$ and a specifically selected retrieval strategy. The retriever takes an input query

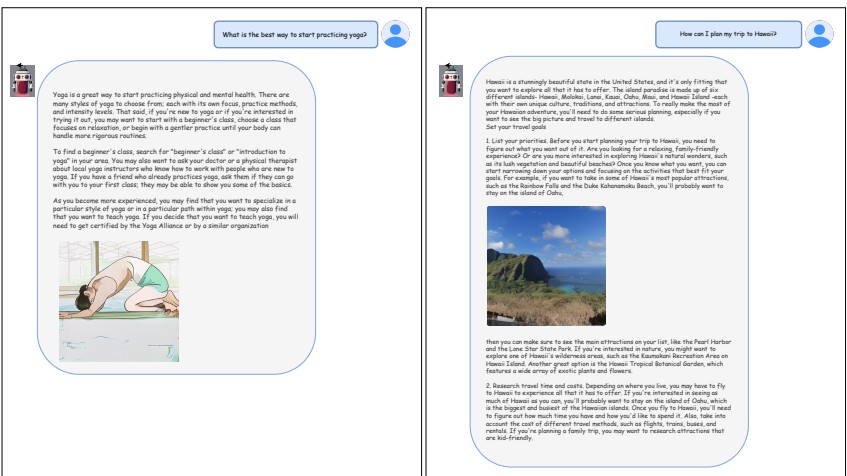

Figure 3: Samples generated by our JAM-Cross instruct tuned model. (Left - generated without retrieval augmentation; Right - generated with retrieval augmentation)

$x$ and returns a relevance score $r(x, m)$ for each candidate document $m \in \mathcal{M}$. Each multimodal document is split between text and images and fed to the corresponding modality-specific VIT-B-32 CLIP encoder (Radford et al., 2021). The two embeddings are then averaged to form the documents' vector representation. We then use Maximum Inner Product Search (MIPS) over the memory bank to obtain a list of candidates. When sampling retrieved documents, we prioritize the diversity of the sampled documents by skipping candidates with a score $r(x, m) \geq 0.9$. We apply query dropout to regularize the training, dropping 20% of tokens from the input sequence $x$.

**Training - Alignment Phase**   During the continued pretraining, we train for approximately 50B multimodal tokens. Our initial learning rate is $lr = 3 \times 10^{-5}$ we use 500 warm-up steps. We set our optimal batch size to 8M tokens, this hyperparameter is borrowed from the mixed-modal scaling laws introduced in Aghajanyan et al. (2023). The total number of training steps is 5960. This training procedure takes approximately one day on 256 80GB A100s for all models. We select the last checkpoint for all the different JAM models, which is always the one with the lowest average validation perplexity (PPL).

**Training - Instruct Tuning**   Our instruct tuning training procedure is data efficient we train with our instruction tuning mixed corpora. The initial learning rate is set to $1 \times 10^{-5}$, and we use 300 warm-up steps and a batch size of 1M. The instruction tuning procedure takes less than 2 hours on 64 80GB A100s, we train for 15 epochs over our mixture of datasets and manually select the best checkpoint corresponding to the 9th epoch. Following Zhou et al. (2023), we notice that the validation PPL doesn't correlate with the quality of the responses.

**Decoding Strategies**   We implement a mixed-modal decoding strategy for our interleaved generation. The model starts generating text tokens until a modality <break> token is detected, then an image is sampled. The generation process alternating the two modalities continues iteratively until a <eos> token is sampled. As a result our model is able to generate free-form multimodal documents. We employ temperature sampling, a common technique used in autoregressive model (e.g Ramesh et al. (2022)) to control the randomness of the prediction by modifying the softmax temperature $\tau$. We pair this technique with TopP sampling introduced in Holtzman et al. (2019) consisting of sampling from the top-ranked tokens with a cumulative probability exceeding a predefined threshold $\tau_P$. We also employ classifier-free guidance (CFG (Gafni et al., 2022)) for sampling images. This technique allows to condition the sampling procedure, blending the logits from an unconditional sample with the logits from a conditional sample. The procedure is mathematically described as

$$\text{logits}_{cf} = \text{logits}_{uncond} + \alpha_c(\text{logits}_{cond} - \text{logits}_{uncond}) \tag{6}$$

where $\text{logits}_{cond} = \mathcal{T}(t_y|t_x)$ and $\text{logits}_{uncond} = \mathcal{T}(t_y| <mask>)$; $\mathcal{T}$ represent the transformer model, $<mask>$ represent the absence of the input text, $t_x$ are the conditional input tokens, $t_y$ are

the output tokens and $\alpha_c$ is the scaling factor for CFG. Thanks to the CM3 objective, our training procedure allows our models to sample with CFG without further fine-tuning. Inspired by Yu et al. (2023) we complement this technique to boost the generation quality. Our samples are generated using a temperature value $\tau = 1$, $\tau_P$ is set between $0.8$ and $1$, and we use classifier-free guidance with values $3.5$ and $4$. In contrast to other approaches, we don't make use of the computationally expensive clip-reranking (Ramesh et al., 2021; Yu et al., 2022; Gafni et al., 2022) or constrastive decoding (Li et al., 2022b; Yu et al., 2023).

### 3.1.1 DATASETS

**Shutterstock** We randomly sample a subset of 30B tokens from CM3leon (Yu et al., 2023) pretraining data. The data consists of legally acquired image-caption pairs from Shutterstock, a commercial online platform offering images with ownership attribution and clear licensing terms.

**Text corpora** We use 30B text tokens sampled from a mixture of several publicly available data, and we reuse the data used for training other common open-source LLM following the same preprocessing of (Touvron et al., 2023). The datasets are: English CommonCrawl (Touvron et al., 2023), C4 (Raffel et al., 2020), Wikipedia, Books3 from ThePile (Gao et al., 2020), and arXiv.

**LIMA** We use the 1k dataset present in Zhou et al. (2023), which features various curated prompts and responses.

**wikiHow** We collect an interleaved image-text dataset sampling 3000 articles from WikiHow, an online wiki publication that usually curates apposite images for each article. We sample balanced articles from each category to ensure diversity; moreover, we leverage the platform's community ratings to filter each article's quality, sampling only those with a score greater than $90/100$. For each article, we use the title (e.g., *'How to make ..?'*) as prompt, we modify the phrase *'This article...'* with *'The following answer..'*. Furthermore, we restrict the number of images as 3 per sample, to fit our 4096 context length.

### 3.2 CONTINUED PRETRAINING RESULTS

In the initial stage of continued pretraining, we evaluate the performance across various JAM models. Our primary objective is to ensure minimal performance degradation post-merging, relative to the parent models. Managing both image and text processing within a single model poses significant challenges. This evaluation seeks to quantify the retention of original performance in our different JAM models, benchmarked against the two parent models specialized in individual modalities.

### 3.2.1 TEXT MODALITY

For the text modality, we compare the zero-shot performance on some common sense reasoning tasks: PIQA (Bisk et al., 2020), ARC-Challenge, ARC-Easy (Clark et al., 2018), StoryCloze (Mostafazadeh et al., 2016), Winograd, and Winogrande (Sakaguchi et al., 2021). We also report some recent influential LLM (Brown et al., 2020; Touvron et al., 2023), and our LLM (Molybog et al., 2023) fine-tuned with 4k context as a reference. Results are presented in Table 1. The JAM-Uniform reaches slightly better text-only performance than JAM-Width however, it is crucial to remark that this approach consolidates the functionalities of both parent models within a constrained 7B parameter space. Our findings reveal that the intrinsic knowledge of the parent models can be recovered mainly from the parameter average utilizing only a minimal portion of the original pretraining data. The JAM-Cross model yields the best results, aligning with our primary LLM. This highlights the strength of our bidirectional cross-attention mechanism against other baselines.

### 3.2.2 IMAGE-TEXT MODALITY

To assess the performance of our different baselines over the image-text modality, we compare them using the validation perplexity (PPL) on MS-COCO dataset (Lin et al., 2014). We believe this metric robustly correlates with performance on subsequent tasks, such as image generation and captioning. Furthermore, it provides a reliable reference point for comparing different autoregressive models sharing an identical tokenizer. Results are reported in Table 2. Diverging from results

Table 1: Zero Shot Text Comparison on Common Sense Reasoning Tasks

| Model | Size | PIQA | ARC-C | ARC-E | StoryCloze | Winograd | Winogrande |
|-------|------|------|-------|-------|-----------|----------|------------|
| GPT-3 | 175B | 81.0 | 51.4 | 68.8 | - | - | 70.1 |
| LLaMa | 7B | 79.8 | 47.6 | 72.8 | - | - | 70.1 |
| LLM-4k | 7B | 76.7 | 45.9 | 67.7 | 79.3 | 83.9 | 66.2 |
| JAM-Uniform | 7B | 62.4 | 28.5 | 42.6 | 63.5 | 47.8 | 49.7 |
| JAM-Width | 26B | 57.8 | 31.4 | 31.6 | 54.7 | 50.2 | 51.9 |
| JAM-Cross | 19B | 75.4 | 41.6 | 67.2 | 79.8 | 81.0 | 66.0 |

Table 2: Image-Text Comparison

| Model | Size | MS-COCO PPL |
|-------|------|-------------|
| CM3 | 2.7B | 200.1 |
| RA-CM3 | 2.7B | 193.1 |
| CM3leon | 760M | 168.8 |
| CM3leon | 7B | 149.0 |
| JAM-Uniform | 7B | 177.5 |
| JAM-Width | 26B | 159.5 |
| JAM-Cross | 19B | **147.6** |

Table 4: Ablations - JAM-Cross Model

| C-Attn | Size | Wikipedia PPL | MS-COCO PPL |
|--------|------|---------------|-------------|
| ✗ | 13B | 7.86 | 153.2 |
| 1 | 26B | 7.53 | 152.4 |
| 2 | 19B | **7.18** | **149.0** |
| 4 | 16B | 8.55 | 151.7 |

Table 3: Ablations - JAM-Width Model

| Init. | Wikipedia PPL | MS-COCO PPL |
|-------|---------------|-------------|
| Copy | **7.34** | **159.5** |
| Average | 9.0 | 175.4 |

Table 5: Ablations - Instruction Tuning

| Shutterstock | MS-COCO PPL |
|--------------|-------------|
| ✗ | 190.2 |
| ✓ | 164.5 |

on the text-only modality, the JAM-Width model exhibits enhanced performance over the JAM-Uniform model in the image-text domain. Specifically, the JAM-Width model demonstrates superior efficacy in retaining image-text performance relative to text-only performance. Conversely, despite a decline in performance, the JAM-Uniform model remains a good parameters-performance trade-off. Interestingly, our JAM-Cross model not only reaches the best PPL between the JAM strategies but also surpasses our foundational image-text model, CM3leon. We hypothesize that such advancement can be attributed to integrating novel textual capabilities coupled with an augmented parameter count inherent to the combined architecture. Based on empirical evidence, the JAM-Cross emerges as the best strategy to combine two pretrained autoregressive models.

### 3.2.3 INTERLEAVED GENERATION

Our instruct-tuned JAM-Cross model reaches a high-quality level of image-text generated output. To demonstrate its ability to generate coherent modality interleaved responses, we show an extensive set of generated samples in Figure 3 and Section B. The samples generated with retrieval are obtained from the model instruct-tuned with a mixture of pretraining image-text Shutterstock data along with our corpora of instruct-tuning datasets, while the samples generated without retrieval are obtained from the model instruct tuned only on our instruct-tuning set. The generated samples show coherent image and text integration, demonstrating unprecedented abilities at this novel task. Overall we find our retrieval augmented solution to be more effective than standard image sampling, boosting image quality. We further report several qualitative comparisons with the most relevant previous work GILL (Koh et al., 2023a) that features mixed-modal generation. We use our retrieval-augmented JAM-Cross model and source generations for the GILL model from the original paper. From this comparison (Figure 4), it's immediate to notice how our model has a better overall quality of responses. The generated text is more complete and exhaustive, while the generated images are more relevant to the text context. We remark that our method is the first capable of such coherent and interleaved generation with a focus on instruction tuning and that our fine-tuning procedure is effective in

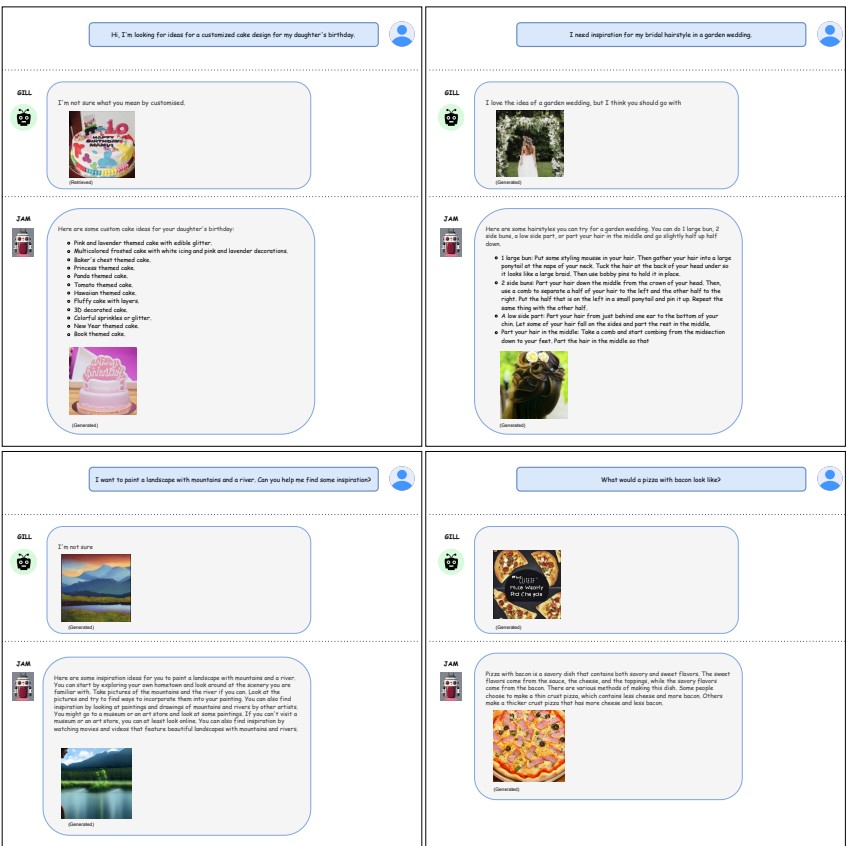

Figure 4: Qualitative comparison with previous interleaved generation models. Compared to GILL, our model is able to generate more complete and precise answers. Results for GILL are sourced from Koh et al. (2023a).

efficiently learning the style of the dataset, not only for text but even for images. Our model paves the way toward a larger adaption of mixed-modal generation in real-world use cases.

## 3.3 ABLATION STUDY

We compare the two approaches for the width concatenation model: copying the original models' weight or using the average to initialize the new parameters. Results (Table 3) show that copying the weights is more effective than averaging them to retain the original model capabilities. The ablation study for the Cross-attention model is presented in Table 4. We ablate the frequency of inserting cross-attention layers and the impact of not using any cross-attention layers. These experiments are performed training with 25B tokens, all the other parameters are the same as reported in Sect. 3.1. We remark that this is an even shorter training setting concerning our 50B tokens total training and that the difference in performance increases as the training progresses. We further ablate the contribution of image-text pretraining data in the instruction tuning procedure in Table 5. The results indicate the importance of using pretraining data mixed in the instruction tuning procedure to preserve the MS-COCO PPL. We do not report WikiHow PPL since analyzing the models shows that it doesn't correlate with generation quality similarly to Zhou et al. (2023).

## 4 RELATED WORKS

**Generative Text-to-Image Models** The field of generative text-to-image models has recently been dominated by diffusion models (Sohl-Dickstein et al., 2015; Ho et al., 2020). Recent enhancements have used pretrained text representations (Ramesh et al., 2022; Nichol et al., 2022) like CLIP (Radford et al., 2021) to improve the generation quality. Concurrently to developing diffusion-based generative models, significant steps have been made by autoregressive token models (Esser et al., 2021; Gafni et al., 2022). These models encode images into a discrete latent space (Van Den Oord et al., 2017)

and can be processed as a standard sequence-to-sequence modeling task, enabling the borrowing of techniques used from Large Language Models. A critical element that has been found beneficial in boosting text-to-image generative models is retrieval augmentation (Chen et al., 2022; Yasunaga et al., 2022). Yasunaga et al. (2022) propose to prefix decoder-only models, such as Aghajanyan et al. (2022), with retrieved images during training, resulting in a huge efficiency gain for the training procedure. Yu et al. (2023), scale this strategy to reach state-of-art performance in image generation using 5x less training compute. In this work, we borrow their model as our text-to-image autoregressive backbone.

**Multimodal Language Models**   The multimodal language model field has recently seen considerable development. Several prior works have focused on connecting language models to visual encoders. (Tsimpoukelli et al., 2021; Mokady et al., 2021; Najdenkoska et al., 2023; Li et al., 2023). These methods typically train a mapping network between a pretrained image encoder and a language model. Flamingo (Alayrac et al., 2022) introduces cross attention into a frozen LLM to inject visual features and trains a large corpus of image-text pairs. In this work, we similarly use cross attention to bridge the two models; however, our mechanism is bidirectional between the vision and language models, while for Flamingo, the visual knowledge is injected in the language model and not vice-versa. CM3 (Aghajanyan et al., 2022) is trained on a large corpus of structured HTML; it introduces the Casually Masked Language Modeling objective we adopt to train our models. Koh et al. (2023b) propose a multimodal language model capable of processing arbitrarily interleaved image and text inputs and generating interleaved output of text and retrieved image. Subsequently, on the same line of work, GILL Koh et al. (2023a) proposes to ground an LLM to a text-to-image model, using a mapping network and freezing the pretrained models, introducing the possibility of generating or retrieving images as output. Similarly to GILL, Sun et al. (2023b) propose to model different modalities in an autoregressive way with a single model they call Emu. Differently from our work, they employ EVA-CLIP (Sun et al., 2023a) encoder to generate visual embeddings and Stable Diffusion (Rombach et al., 2022) conditioned on the generated image tokens to decode images.

**Instruction Tuning**   Instruction tuning aims to teach language models to follow natural language instructions. Several methods have been proposed for instruction tuning, using existing NLP datasets converted in instruction formats Wei et al. (2021) Chung et al. (2022), or using LLMs like GPT-4 to generate instruction data with better diversity Wang et al. (2022) Honovich et al. (2022). Recently, LIMA Zhou et al. (2023) demonstrated that 1,000 carefully curated samples are enough to reach competitive results compared to bigger instruction-tuning datasets. The authors hypothesize that most of the knowledge is learned during the pretraining, and the instruction tuning teaches the style to interact with the users. In this work, we explore using a small set of multimodal instruction tuning data to fine-tune our model, verifying the effectiveness of a small dataset in this multimodal setting tailored to image generation. Several vision language works adopt instruction tuning for multimodal tasks-focused user interactions optimized for visual content understanding Liu et al. (2023) Dai et al. (2023) Ye et al. (2023) Zhu et al. (2023). Unlike previous works, we explore instruction tuning focused mixed-modal generation, paving the way for more significant adaptation of multimodal models that can generate interleaved image-text output.

## 5   CONCLUSIONS

In this work, we have presented novel methodologies for combining pretrained autoregressive models, demonstrating the viability of synthesizing the knowledge of two distinct models into a cohesive structure with extended capabilities. Our exploration validates that the integrated model can be adeptly fine-tuned using our tailored instruction-tuning procedure for interleaved image-text generation. To this end, we pioneered creating a specialized dataset centered on instruction tuning for this particular task. Nevertheless, the proposed study is limited to 7B parameter models with the same architecture. Future works may consider scaling the models' size and asymmetrically applying our cross-fusion method to bridge models of varying sizes. Increasing the context length and delving into multi-turn conversations could further represent an interesting exploration direction. In conclusion, our study sets the foundation for substantial advancements in the realm of multimodal autoregressive models. The fusion of text-to-image generation with large language models paves the way for sophisticated systems capable of interleaved image-text interactions, enriching the landscape of conversational AI.

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

## A  LIMITATIONS

The generation quality of our proposed model presents some limitations. Our JAM frameworks rely on LLMs and text-to-image autoregressive models, inheriting their strengths and limitations, such as the potential for hallucinations and biases in image generation. These limitations may be addressed by improving and leveraging better autoregressive backbones.

Moreover, our instruct-tuning procedure focuses on a specific wiki-style single-turn question answering. Most of the time, the model generates a single or, at most, two images, while a longstanding goal could be to generate entire multimodal documents with >2 images. This is partially bottlenecked by the training data and the context length of 4k tokens; given that each image takes 1024 tokens, generate longer outputs it's still challenging.

Furthermore, our model is more effective in generating interleaved image text when the question style aligns with the dataset style and when the images refer to concrete objects. While the model usually struggles to represent more abstract concepts. We showcase some examples of this behavior in Figure 5. When the subject of the text dialogue are more abstract concepts such as "time travel", "music", or "research", the generated images fail to capture the textual context and enrich the model output faithfully.

## B  GENERATED EXAMPLES

This section shows several samples generated by our instruction-tuned models using both retrieval augmentation and standard image generation.

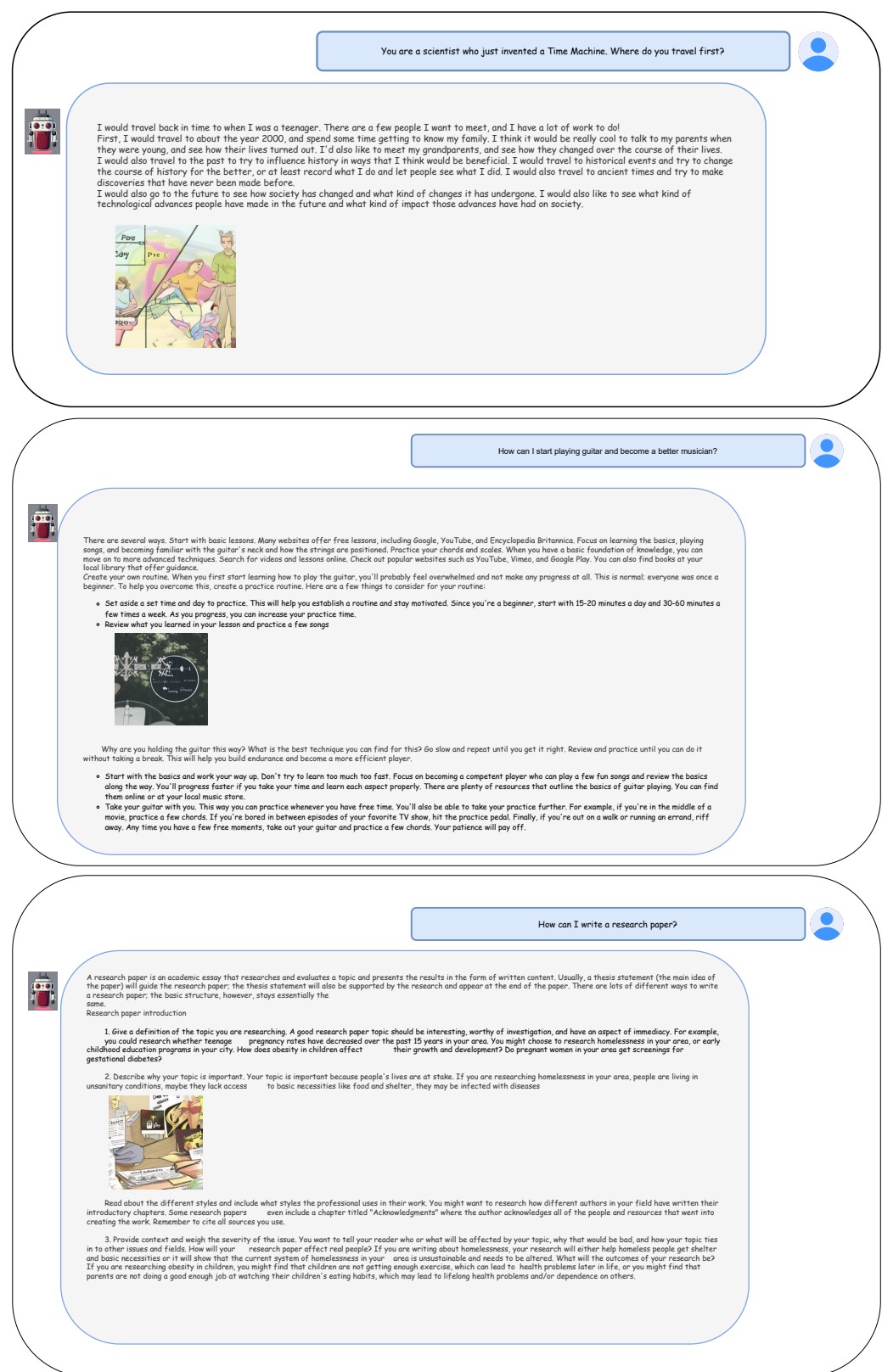

Figure 5: Generated samples - Failure Cases

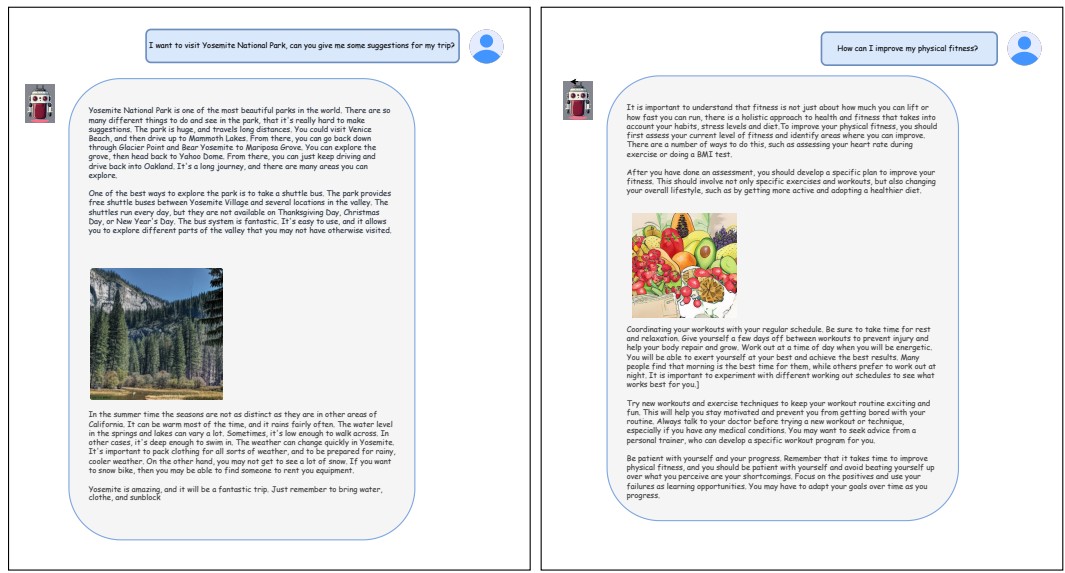

Figure 6: Generated samples - without retrieval (left), retrieval augmentation (right).

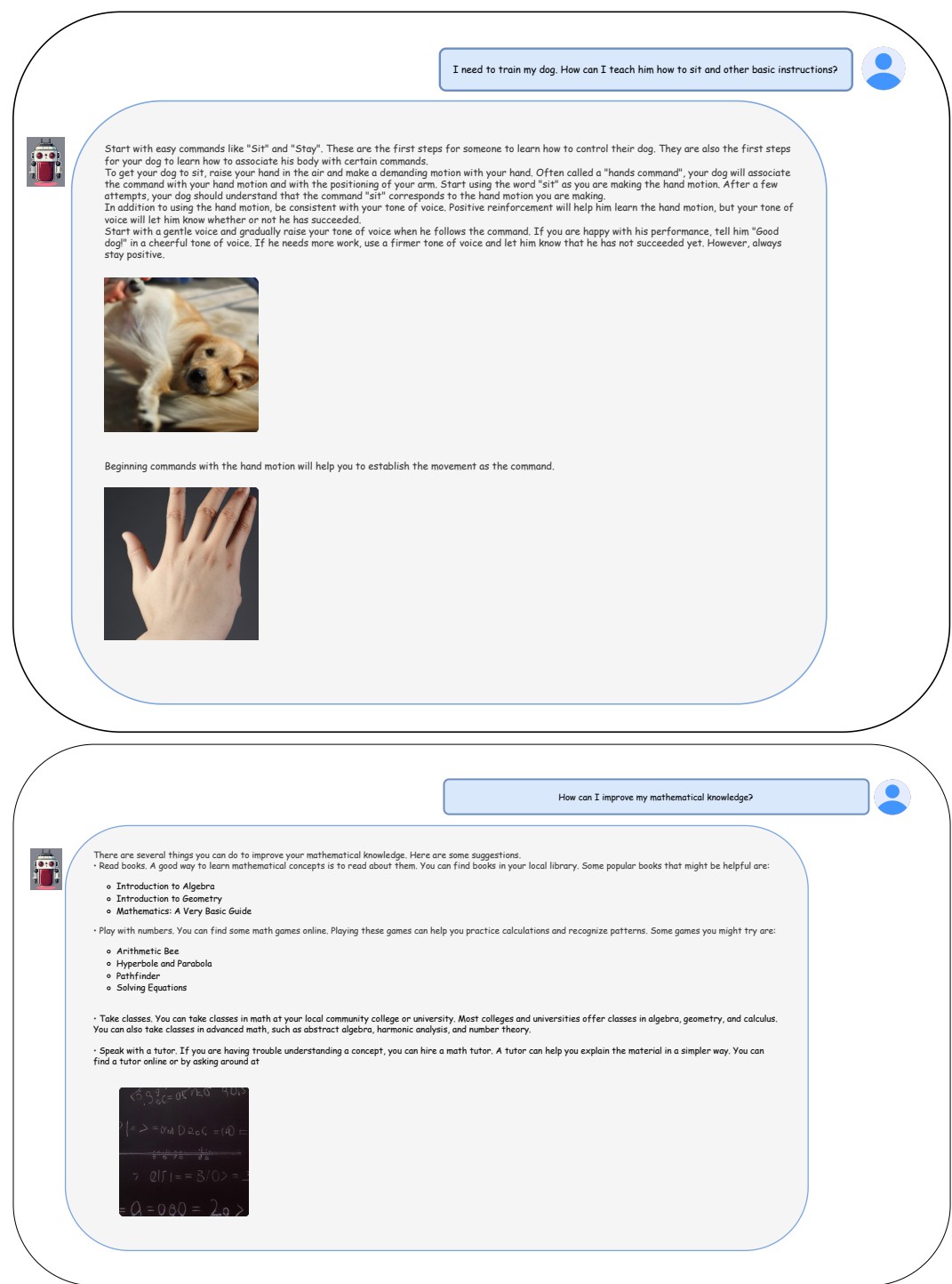

Figure 7: Generated samples - retrieval augmentation

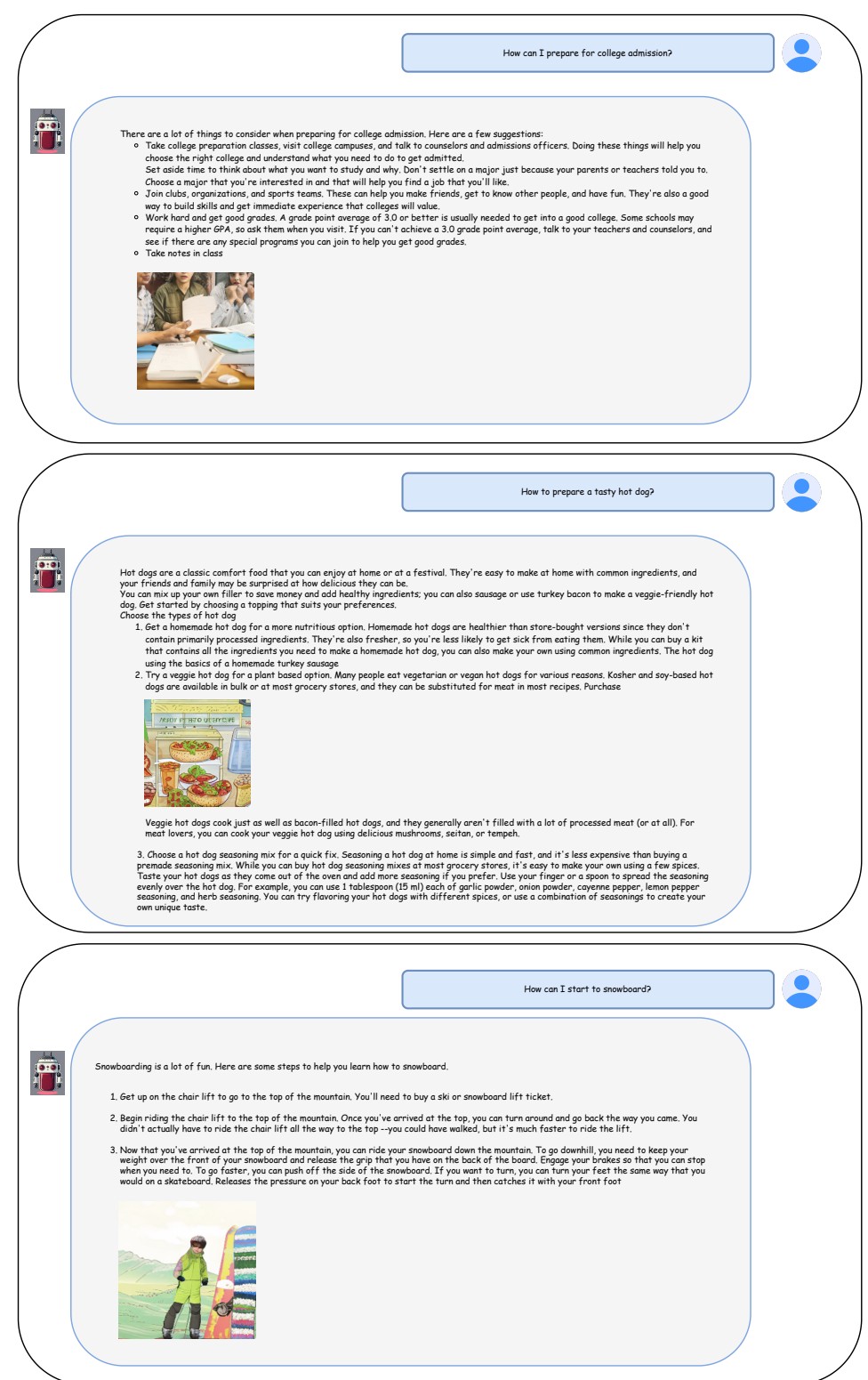

Figure 8: Generated samples - without retrieval augmentation

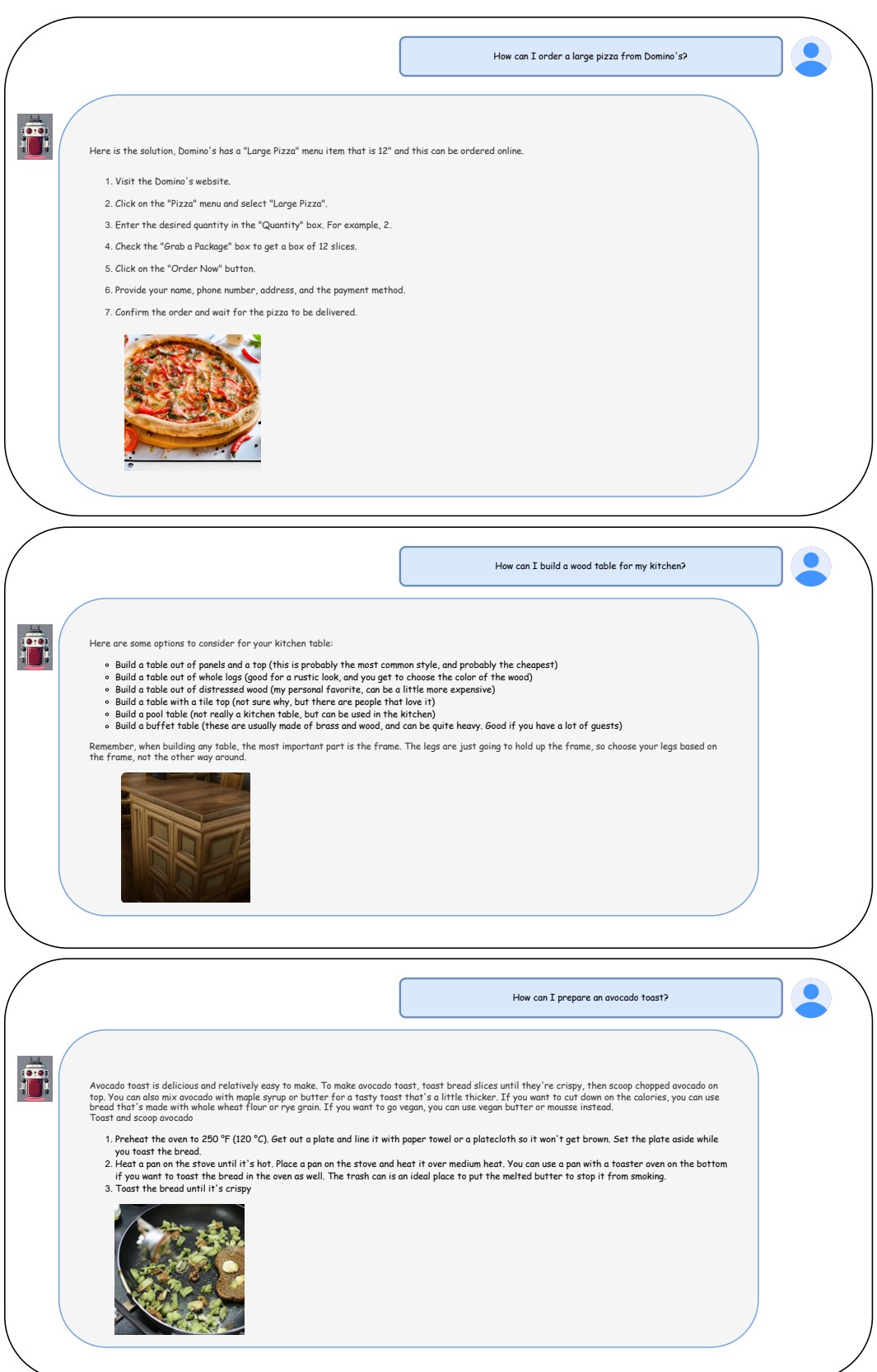

Figure 9: Generated samples - without retrieval augmentation

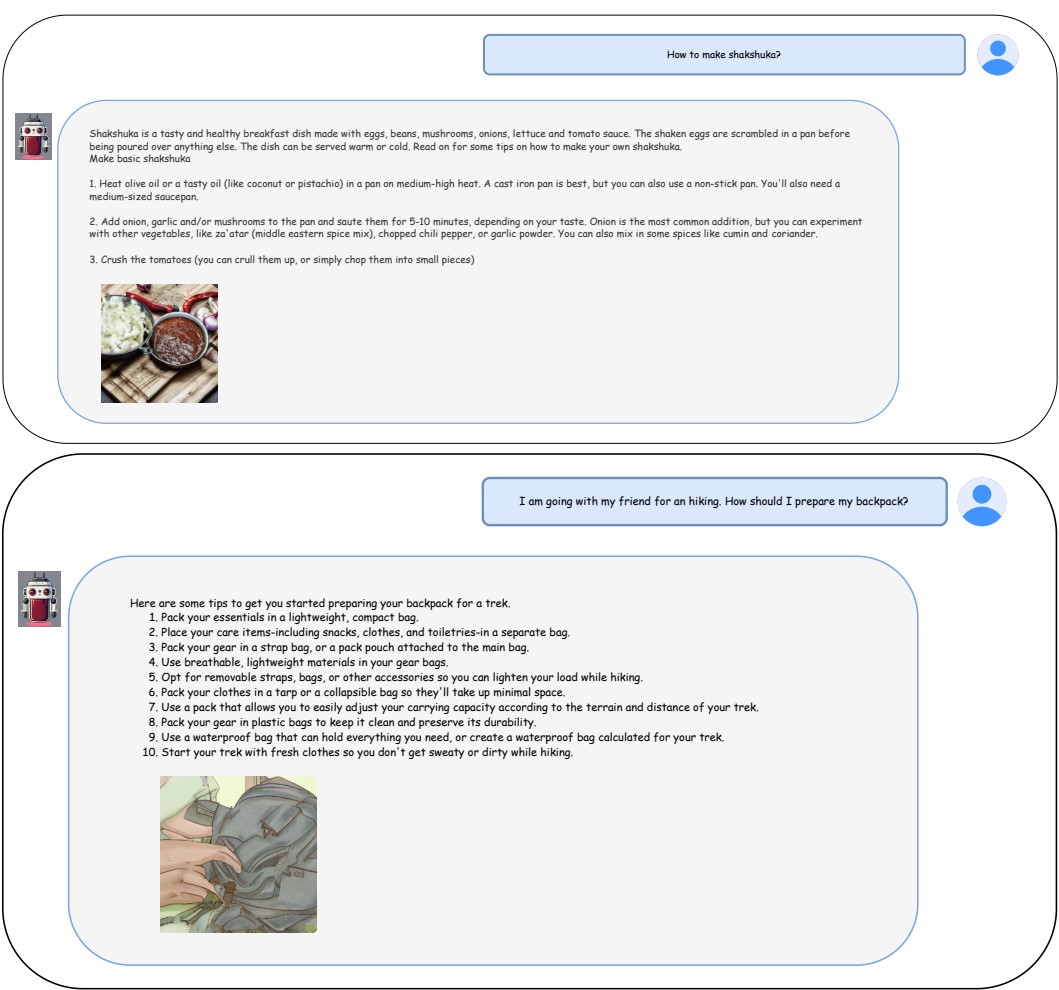

Figure 10: Generated samples - without retrieval augmentation

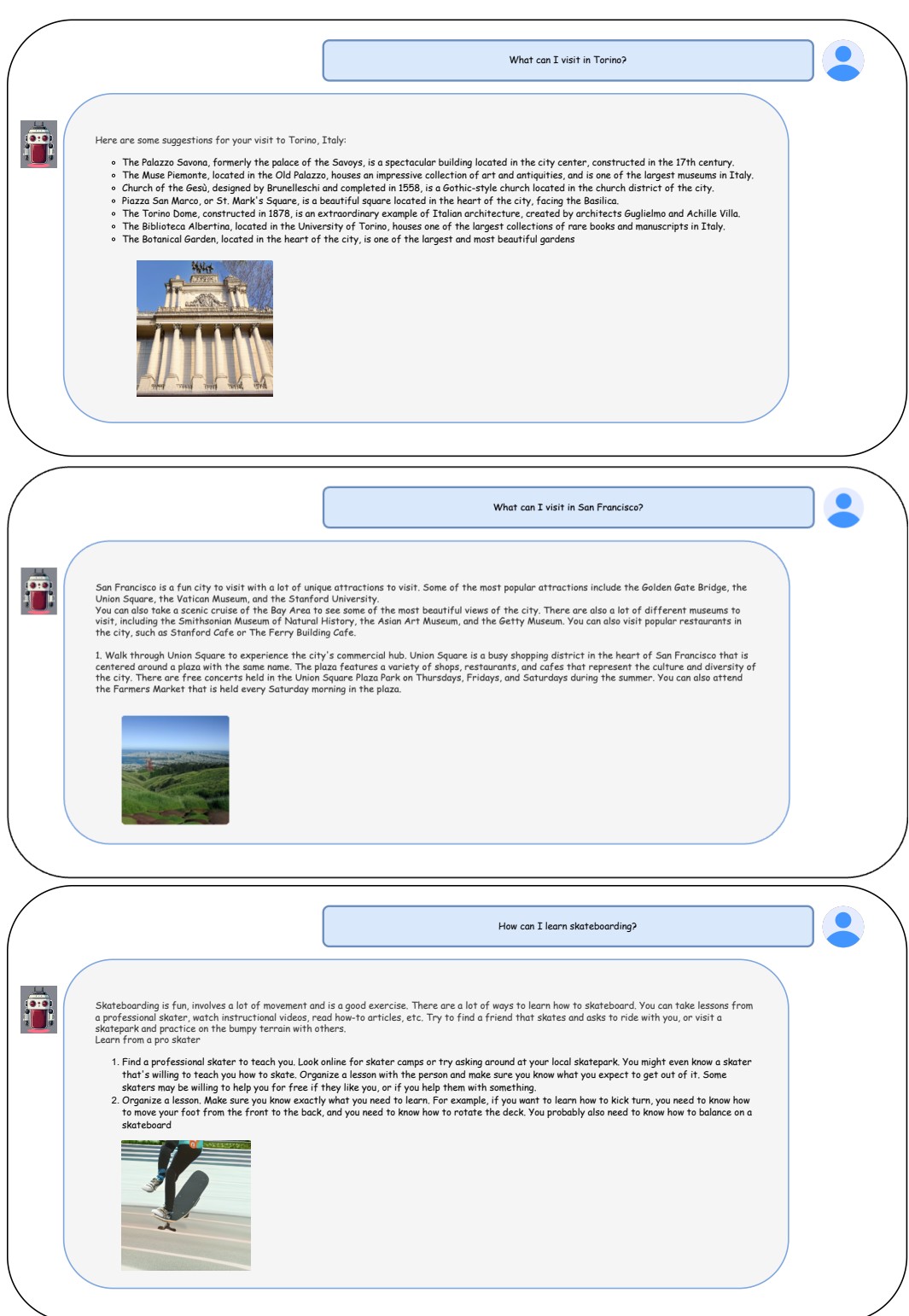

Figure 11: Generated samples - retrieval augmentation

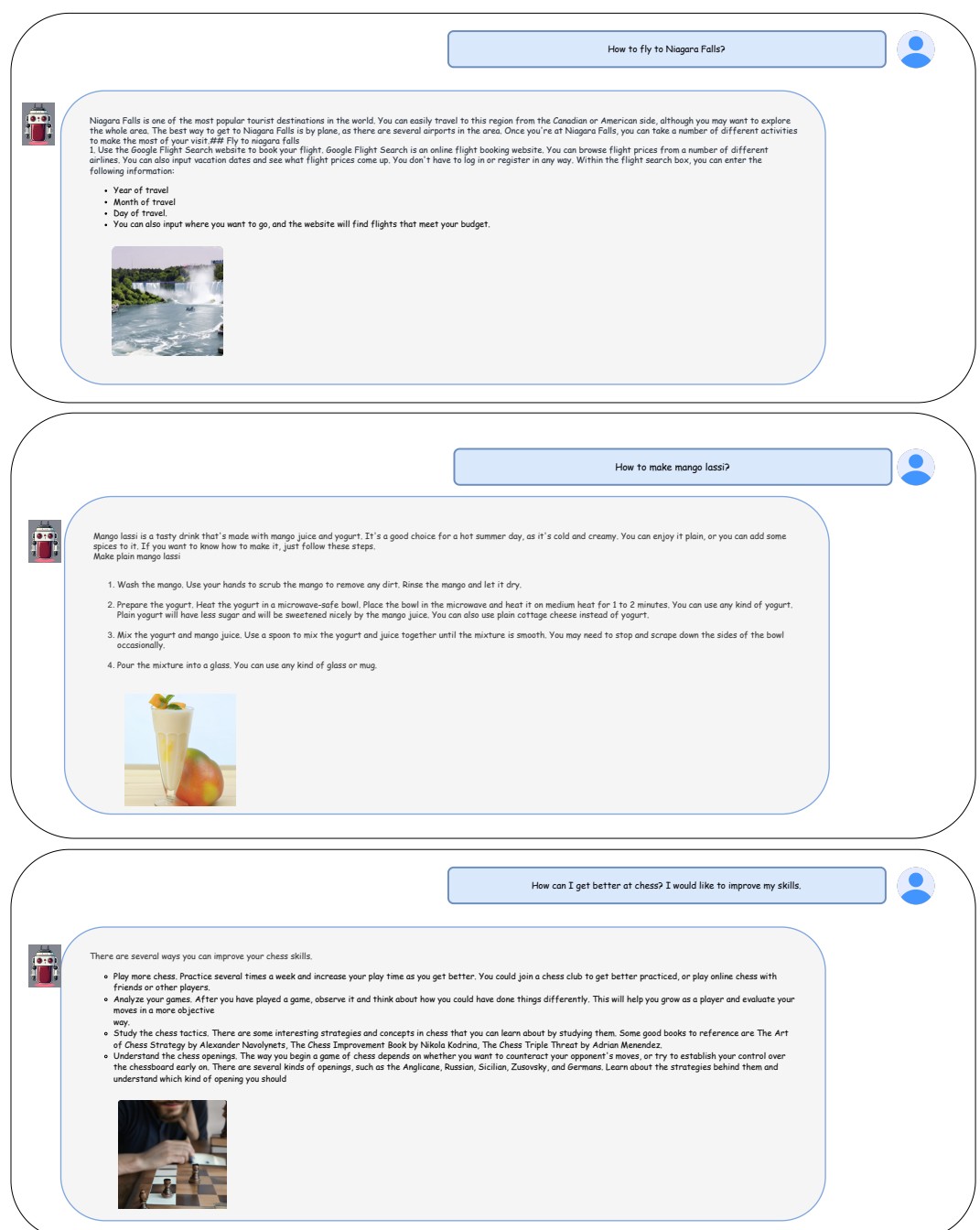

Figure 12: Generated samples - retrieval augmentation

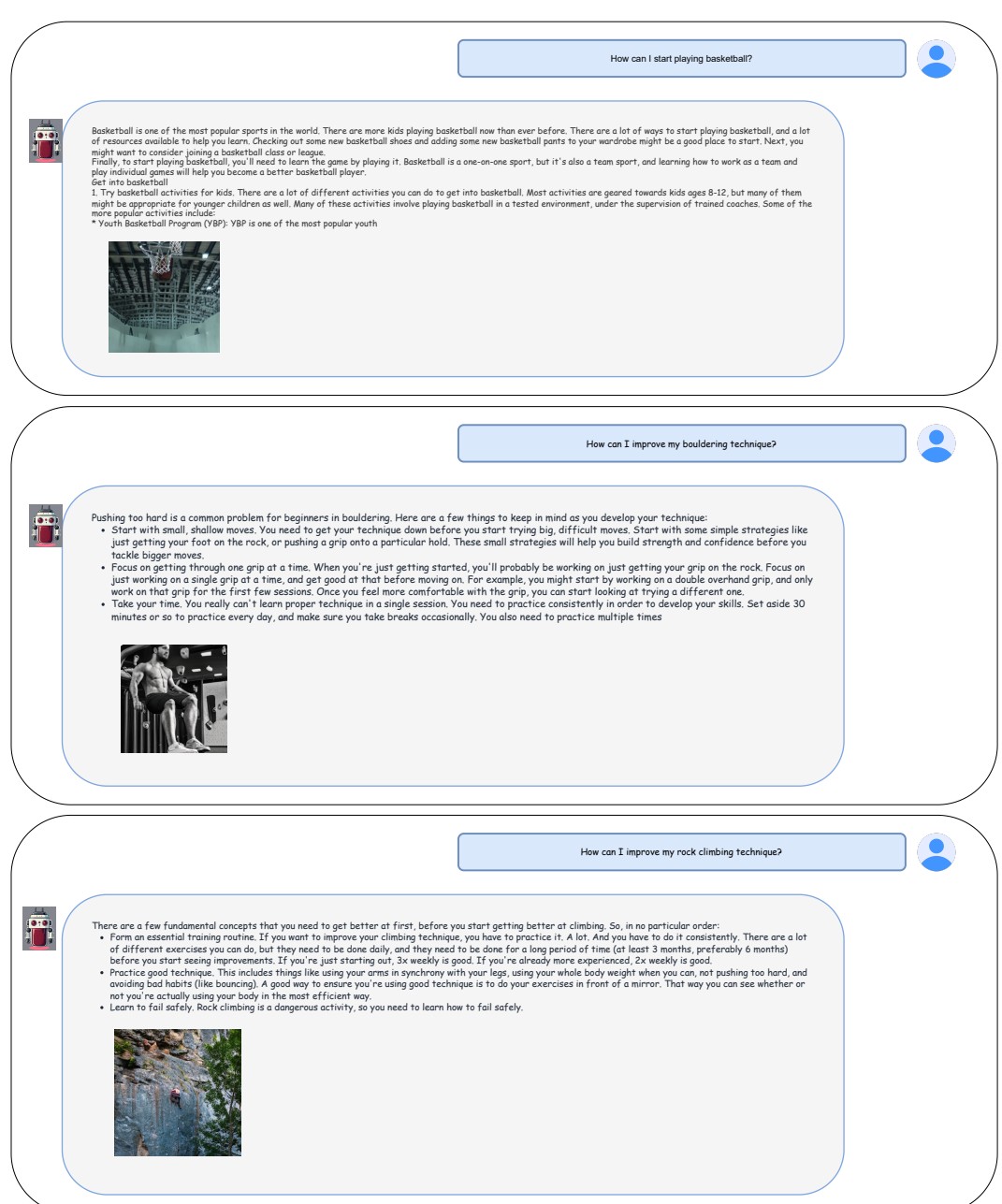

Figure 13: Generated samples - retrieval augmentation

