# OpenReview forum: "Jointly Training Large Autoregressive Multimodal Models"
_ICLR.cc/2024/Conference — ICLR 2024 poster_

### Official Review · Reviewer_eaYf · 2023-11-01

**Soundness:** 2 fair
**Presentation:** 2 fair
**Contribution:** 4 excellent
**Rating:** 6
**Confidence:** 3

**Summary:**

The paper introduces the Joint Autoregressive Mixture (JAM) framework, aimed at integrating large-scale pre-trained text and image models to generate cohesive multimodal outputs. Leveraging the architectural compatibility of autoregressive text-to-image models with Large Language Models (LLMs), the authors propose a systematic approach to model fusion and joint training. The JAM framework exhibits superior performance in generating high-quality multimodal outputs, integrating text and images seamlessly, and represents a significant step towards advanced multimodal conversational systems.

**Strengths:**

- The authors have proposed novel methods for combining pretrained AR models.
- This represents a great contribution towards building multi-modal conversation agents.
- The specialized instruction-tuning strategy demonstrates efficiency and effectiveness.
- The results shown are noticeably better.

**Weaknesses:**

- PPL is not defined anywhere in the draft.
- Retrieval augmented presentation, the last section of page 4, can be improved. a small schema can be helpful.
- Instruction tuning lacks enough details. For example, in the introduction, the efficiency for even 1% of original pretraining data is stated but not mentioned anywhere else in the draft.

**Questions:**

- Why does LLaMA have blanks in Table 1?
- Is Table 1 suggesting that 7B LlaMa is better than all the proposed JAM models in reasoning?
- Can we have the baseline results in Table 2 for the Wikipedia dataset as well?

---

> ### Author Response · Authors · 2023-11-17
> **Response to reviewer eaYf**
>
> We thank the reviewer for the insightful suggestions and valuable feedback, we address the different highlighted weaknesses and questions separately:
>
> > PPL is not defined anywhere in the draft.
>
> You are correct in pointing out that PPL is not defined anywhere in the draft. We will fix this in the revised version of the manuscript to improve clarity of the presentation
>
> > Retrieval augmented presentation, the last section of page 4, can be improved. a small schema can be helpful.
>
> We appreciate your suggestion to enhance the clarity of the retrieval augmented presentation section. We recognize that a visual representation can significantly aid in the comprehension of the model's architecture and its retrieval mechanism. However we omitted it due to space constraints. We plan to improve the clarity of the procedure description in our revised version.
>
> > Instruction tuning lacks enough details. For example, in the introduction, the efficiency for even 1% of original pretraining data is stated but not mentioned anywhere else in the draft.
>
> We appreciate your feedback on the need for more detail in the instruction tuning section of our paper. Our instruction tuning procedure is briefly described in Section 2.2 due to space limitations. We extend the discussion of this phase in the experimental details where we detail the training procedure and describe our WikiHow instruct tuning dataset creation and structure. We recognize the importance of elaborating on this critical aspect of our work and will expand this section in our revised manuscript.
> Regarding the reference to using just 1% of the original pretraining data in the introduction, we aim to clarify this in the revised version. This statement was intended to emphasize the efficiency of our model during the continued pretraining phase. Specifically, the 1% figure pertains to the amount of data used in aligning the two models, which amounts to 50 billion tokens. We will include this clarification in the updated version of the paper.. By doing so, we hope to underscore the data efficiency aspect of our method more prominently, illustrating how our procedure can be effective with a relatively small dataset.
>
> > Why does LLaMA have blanks in Table 1?
> In Table 1, there are some blank entries under the LLaMa model, because it was not assessed in the original paper on certain benchmarks we included in our test. However, it's important to note that these particular benchmarks are not central to our primary comparative analysis.
>
> > Is Table 1 suggesting that 7B LlaMa is better than all the proposed JAM models in reasoning?
>
> We acknowledge that our models, including JAM-Uniform, Width, and Cross, exhibit comparatively lower performance with respect to other baselines like GPT-3 and Llama 7b. This is partly due to our primary focus being on the integration and interplay of different modalities, rather than improvements on reasoning and understanding. It is worth noting that the reasoning performance of the JAM models is intrinsically linked to the capabilities of the initial language model we build upon. Thus, employing a more powerful language model (e.g Llama 7B/13B/70B) as the foundation model could inherently enhance the reasoning abilities of our combined model. This approach aligns with our method, which is designed to preserve and potentially amplify the intrinsic capabilities of the underlying models. Our current focus has been on developing a robust methodology for combining architectures pretrained on different trillion scale datasets, the goal is not to improve text reasoning/understanding, but to leverage the strong pretrained knowledge in our initial backbone.
>
> > Can we have the baseline results in Table 2 for the Wikipedia dataset as well?
>
> We acknowledge the importance of including baseline results for the Wikipedia dataset in our comparison and appreciate your suggestion to add these to Table 2. In the revised version of our manuscript, we will incorporate the following updated table to provide a comprehensive overview:
>
> | Model             | Wikipedia PPL |
> |--------------------|----------------------|
> | LLM-4k           |          6.04         |
> | JAM-Uniform |          9.06         |
> | JAM-Joint        |          7.34         |
> | JAM-Cross      |          6.95         |
>
>
> The other models, including CM3, RA-CM3, and CM3leon, are exclusively image-text models. As a result of their training, which predominantly involves a limited amount of text tokens, typically in the form of image captions, they do not perform as well in terms of Wikipedia PPL scores.
>
> We thank the reviewer for the suggestions that helped us improve the quality and clarity of the paper.

---

> ### Author Response · Authors · 2023-11-22
> **Follow up on response**
>
> Dear reviewer eaYf,
>
> We would like to bring to your attention that tomorrow (Nov 22) marks the conclusion of the rebuttal period. We are eager to know if our response has sufficiently addressed your concerns, or if there are additional questions you may have. We value the chance to engage in further discussion should there be any unresolved issues. In case our response has met your expectations, we kindly request that you consider modifying your initial score. Thank you once again for your attention and time.
>
> Warm Regards

---

### Official Review · Reviewer_Ghzd · 2023-11-01

**Soundness:** 3 good
**Presentation:** 3 good
**Contribution:** 2 fair
**Rating:** 5
**Confidence:** 3

**Summary:**

In this work, the authors study three ways in which a pre-trained text-only decoder LLM $\mathcal{T}\_{llm}$ and a text-image LLM (trained on VQ-VAE tokens) $\mathcal{T}\_{img}$ with the same architecture can be merged as a multi-modal LLM capable of interleaved text-image generation called Joint Autoregressive Mixture (JAM). The three ways of combining the two models are: 1. **Model Merging** in which the weights $\theta\_{llm}$ of $\mathcal{T}\_{llm}$ and the weights $\theta\_{img}$ of $\mathcal{T}\_{img}$ are averaged as initialization weights $\theta\_{avg} = 1/2 \theta\_{llm} + 1/2 \theta\_{img}$ for a model of the same parameter size (7B in the paper) (JAM-Uniform) 2. **Width Concatenation** in which the weight matrices are concatenated to form matrices of double size (per matrix dimension), both copying and averaging (JAM-Width) 3. **Cross Model Fusion** in which both  $\mathcal{T}\_{llm}$ and $\mathcal{T}\_{img}$ are augmented with cross-attention layers that permit a bi-directional flow of information, generalizing the Flamingo model (JAM-Cross). In each way, the authors train the resulting JAM model using the CM3 loss and perform instruction tuning using a small curated dataset of examples (following the LIMA paper).

**Strengths:**

- The idea of bi-directional cross-attention layers between two generative backbones is an elegant approach for performing model merging. Indeed, the authors obtained SOTA results with JAM-Cross on MS-COCO (147.6 PPL), also showcasing the positive influence of the pre-trained joint text decoder.
- Comparing the three ways of performing model merging while demonstrating the superiority of JAM-Cross between the three sheds light on what is the best way of performing such an operation.

**Weaknesses:**

- While the experimental section is fair, pointing to a new state-of-the-art over CM3leon, the improvement is only 1.4 PPL points with an increase of the model size of more than double (19B vs 7B of CM3leon) and increased training time. The authors suggest that minimal performance degradation post-merging should be studied in the evaluation part. While I agree minimal degradation is a necessary condition, it is not sufficient: given that the final model performs the same tasks as CM3leon, improving only 1.4 PPL points does not justify the necessity of jointly training with a text LLM.

**Questions:**

- Shouldnt we have $\mathbf{H}\_{llm,l}$ and $\mathbf{H}\_{img,l}$ in Eq. 4 instead of $\mathbf{H}\_{llm,l}$ and $\mathbf{H}\_{llm,l}$?
- "We prevent the objective from masking across the modality `<break>` tokens.": Does the CM3 objective relocate masks at the end of each modality or at the end of the total document?
- What is $q$ in “The retriever takes an input query x and returns a relevance score $r(q, m)$”, the authors intended the $x$ variable?
- “We leverage our text-to-image backbone’s modifications introduced in Yu et al. (2023).” What does this phrase mean in the paragraph?
- “The two embeddings are then averaged to form the documents’ vector representation”: if a document contains multiple text and images, shouldn't it have more than two embeddings?
- "We prioritize the diversity of the sampled documents by skipping candidates with a score r(q, m) ≤ 0.9.”. Reading the original paper, the candidates with a score higher than 0.9 are dropped, not the opposite as here.
- “we is data efficient we train”, bad phrasing.
- In classifier-free guidance, $t_x$ refers to the tokens generated before the image token to be sampled?
- I don't understand if the final text-to

---

> ### Author Response · Authors · 2023-11-17
> **Response to reviewer Ghzd**
>
> We thank the reviewer for the insightful suggestions and valuable feedback, we address the different highlighted weaknesses and questions separately:
>
> > While the experimental section is fair, pointing to a new state-of-the-art over CM3leon, the improvement is only 1.4 PPL points with an increase of the model size of more than double (19B vs 7B of CM3leon) and increased training time. The authors suggest that minimal performance degradation post-merging should be studied in the evaluation part. While I agree minimal degradation is a necessary condition, it is not sufficient: given that the final model performs the same tasks as CM3leon, improving only 1.4 PPL points does not justify the necessity of jointly training with a text LLM.
>
> We appreciate the reviewer's perspective on the performance improvements of our model over CM3leon. It's important to clarify that while our model indeed shows a modest improvement of 1.4 PPL points over CM3leon, the primary aim of our methodology extends beyond just enhancing this metric. Our goal with this evaluation was to facilitate comparisons between our model and other established autoregressive multimodal architectures, including RA-CM3, CM3, and CM3leon. This was done to ensure that the integration of our two backbones does not compromise their individual performance. On the other hand this work focuses on integrating diverse capabilities from different large-scale datasets. The core innovation lies in enabling the merged model to perform interleaved generation, a capability that our backbone, CM3leon, does not inherently possess. While CM3leon is adept in image-text tasks (captured by MS-COCO PPL) having been trained on a large corpus of image-text pairs, it lacks the nuanced common sense reasoning that is typically developed through extensive large scale text-only training (captured by common-sense reasoning benchmarks). Our model amalgamates these distinct abilities, thus not only retaining the image-text capabilities of CM3leon but also enriching it with the textual understanding characteristic of large language models (LLMs). The proposed approach results in a model with a broader spectrum of abilities, capable of seamlessly generating both images and text. Therefore, the value of our joint training approach with an LLM lies in creating a more versatile and comprehensive model, which we believe is a significant step forward in the field. The modest PPL improvement, while notable, is just one aspect of the overall advancements our model offers, particularly in terms of its expanded functional capabilities and the new avenues it opens for multimodal chatbots.
>
> > Questions:
> > Shouldn’t we have $H_{llm, l}$ and $H_{img,l}$ in Eq. 4 instead of $H_{llm, l}$ and $H_{llm, l}$?
>
> This equation represents the bidirectional cross-attention mechanism in our model, where the query originates from the language model (LLM) and the key and values are derived from the text-to-image model. Specifically, '$H_{img,l−1}$' in the equation refers to the representation from the image-text model at the $(l-1)th$ layer, which is used in the calculation of '$H_{cross,l}$'.
>
> > "We prevent the objective from masking across the modality <break> tokens.": Does the CM3 objective relocate masks at the end of each modality or at the end of the total document?
>
> In our implementation, as per the CM3 objective introduced by Aghajanyan et al. (2022), masks are relocated to the end of the entire document rather than at the end of each modality. Specifically, our model is designed to enhance its versatility for both infilling and standard autoregressive generation. By relocating masks to the end of the total document, our model can more effectively process and generate content that maintains the integrity of each modality, whether it be text or image. This design decision is aligned with the overarching goal of our model to seamlessly integrate and leverage multimodal data. We will clarify the masking strategy in the updated version of the manuscript. For a detailed explanation of this objective and its implications in multimodal learning, we refer you to the work of Aghajanyan et al. (2022), titled "CM3: A Causal Masked Multimodal Model of the Internet."
>
> > What is $q$ in “The retriever takes an input query x and returns a relevance score $r(q,m)$”, the authors intended the $x$ variable?
>
> We apologize for the oversight, in our revised version, we will correct this inconsistency by consistently using 'q' to represent the input query throughout the paper.
>
> > We leverage our text-to-image backbone’s modifications introduced in Yu et al. (2023).” What does this phrase mean in the paragraph?
>
> This phrase means that we adopt the same modification introduced to the training procedure in Yu et al. (2023) and we successively describe these modifications. We apologize and we will improve the clarity in the revised version.

---

> ### Author Response · Authors · 2023-11-17
> **Response to reviewer Ghzd (pt.2)**
>
> > “The two embeddings are then averaged to form the documents’ vector representation”: if a document contains multiple text and images, shouldn't it have more than two embeddings?
>
> You are correct that if the documents contain multiple text and images we will have more than two embeddings, however this retrieval augmentation procedure is done during the training process for image-text data where we train with image-text captions containing a single image and text entry for each document. We do not apply retrieval augmented training during the instruction tuning phase, where we have more than one image /text per document. We will clarify this in the revised manuscript.
>
> > "We prioritize the diversity of the sampled documents by skipping candidates with a score r(q, m) ≤ 0.9.”. Reading the original paper, the candidates with a score higher than 0.9 are dropped, not the opposite as here.
>
> We apologize for the typo; the skipped candidates are those with score r(q,m) >= 0.9, will ensure this correction is reflected in our revised version of the paper.
>
> > In classifier-free guidance, t_x  refers to the tokens generated before the image token to be sampled?
>
> Exactly, in this case they represent the text tokens preceding the image generation.
>
> We thank the reviewer for these considerations and its attention to the details that helped us improve the clarity and correctness of the presentation.

---

> ### Author Response · Authors · 2023-11-22
> **Follow up on response**
>
> Dear reviewer Ghzd,
>
> We would like to bring to your attention that tomorrow (Nov 22) marks the conclusion of the rebuttal period. We are eager to know if our response has sufficiently addressed your concerns, or if there are additional questions you may have. We value the chance to engage in further discussion should there be any unresolved issues. In case our response has met your expectations, we kindly request that you consider increasing your initial score. Thank you once again for your attention and time.
>
> Warm Regards

---

### Official Review · Reviewer_jvv1 · 2023-11-05

**Soundness:** 4 excellent
**Presentation:** 3 good
**Contribution:** 3 good
**Rating:** 6
**Confidence:** 3

**Summary:**

The authors work on combining two pre-trained autoregressive models to form a new decoder armed with multimodal generation capability. More specifically, the authors combined text-to-image decoder and text decoder, and formed a decoder that can seamlessly generate text+image.

**Strengths:**

To the best of my knowledge, this work is, if not the earliest one, among the pioneering works that explore fusing two decoder-only models of different modality into one, to enable seamless image+text generation. Previous works have considered merging the token spaces from different modalities, and use single decoder to enable generating both modalities (like AudioPaLM); However, the idea of fusing two decoders, and arm the new decoder with the capability of generating high-quality multimodal output is new.

As a pioneering exploration, the authors explored three different approaches to fuse two decoders, and all the three methods all seem to be successful. The authors also conduct ablation studies to better understand the three approaches they have proposed. Empirically, the authors show the feasibility of fusing two decoders from multiple modalities.

The authors collect a small and curated mixed-modal dataset for the purpose of multimodal conversational instruction tuning, and also confirmed that training with interleaved image-text data is beneficial.

**Weaknesses:**

On Reasoning/Understanding:

One weak point is that JAM models (both JAM-Uniform, Width or Cross) are all much weaker compared to LLaMA and GPT-3, while LLaMA is even smaller in terms of model size compared to JAM-Width and JAM-Cross. This is acceptable as this work is not focusing on reasoning and understanding.

On Scaling up:

Compared to GPT-3, the JAM model is still pretty small. Scaling up the model size could potentially help with the performance in terms of common sense reasoning and also generation.

Single image generation:

 Interleaved generation is one innovation of this work. By reading the decoding strategies, it seems that one single image will be sampled if <break> token is detected, and then the model will continue text generation unless the <eos> token is sampled. By looking at all the samples provided, it seems that this is the case.One possibility is that there could be multiple text requests to generate multiple images at once.

**Questions:**

Did authors compare their decoding strategy with contrastive decoding and clip-reranking, in terms of both efficiency and performance?

Can users control alpha_c, tao and other critical hyper-parameters during decoding? So the tool can be more balanced in generating text+image, or more leaning towards text or image generation.

---

> ### Author Response · Authors · 2023-11-17
> **Response to reviewer jvv1**
>
> We thank the reviewer for the insightful suggestions and valuable feedback, we address the different highlighted weaknesses and questions separately:
>
> > On Reasoning/Understanding:
> > One weak point is that JAM models (both JAM-Uniform, Width or Cross) are all much weaker compared to LLaMA and GPT-3, while LLaMA is even smaller in terms of model size compared to JAM-Width and JAM-Cross. This is acceptable as this work is not focusing on reasoning and understanding.
> > On Scaling up:
> Compared to GPT-3, the JAM model is still pretty small. Scaling up the model size could potentially help with the performance in terms of common sense reasoning and also generation.
>
> We acknowledge that our models, including JAM-Uniform, Width, and Cross, exhibit comparatively lower performance in these aspects. This is partly due to our primary focus being on the integration and interplay of different modalities, rather than improvements on reasoning and understanding.
> Regarding the model size, we agree that JAM models are smaller than GPT-3. We recognize that scaling up the model size could indeed enhance its performance, particularly in terms of common sense reasoning and generation capabilities. We consider this an important area for future exploration. It is worth noting that the reasoning performance of the JAM models is intrinsically linked to the capabilities of the initial language model we build upon. Thus, employing a more powerful language model as the foundation could inherently enhance the reasoning abilities of our combined model. This approach aligns with our method, which is designed to preserve and potentially amplify the intrinsic capabilities of the underlying models.
> In summary, while our current focus has been on developing a robust methodology for combining architectures pretrained on different trillion scale datasets, we are enthusiastic about the potential of scaling up our models and exploring more advanced foundational models to push the boundaries in reasoning and understanding in future iterations of our research.
>
> > Single image generation:
> > Interleaved generation is one innovation of this work. By reading the decoding strategies, it seems that one single image will be sampled if <break> token is detected, and then the model will continue text generation unless the <eos> token is sampled. By looking at all the samples provided, it seems that this is the case.One possibility is that there could be multiple text requests to generate multiple images at once.
>
> Thank you for your insightful question regarding the decoding strategy for interleaved generation in our work. We apologize for any confusion caused by the initial explanation. To clarify, when the modality <break> token is detected during the decoding process, our model indeed generates a single image. After this image generation, the model has the flexibility to either generate another modality <break> token, thereby continuing with text generation, or to produce an <eos> token, which concludes the generation sequence. Consequently, the number of text segments and images generated in a single run can vary, depending on the input. This flexibility allows for the generation of multiple images and text paragraphs within the same sequence. For instance, as demonstrated in Figure 1 of our paper, the model can produce sequences with two text paragraphs and two corresponding images. However, it is important to note that in most instances observed, the model tends to predict an <eos> token following the generation of a single image. The model is inherently capable of producing multiple images and text paragraphs in response to varying inputs, as showcased in some of our samples. We will clarify this in the revised version of our paper.

---

> ### Author Response · Authors · 2023-11-17
> **Response to reviewer jvv1 (pt.2)**
>
> > Did authors compare their decoding strategy with contrastive decoding and clip-reranking, in terms of both efficiency and performance?
>
> Regarding interleaved generation we do not try contrastive decoding since our topp generation represents a good trade-off between quality and efficiency. However our model inherits the decoding properties of our backbone CM3leon. The authors of the CM3leon paper [1] propose to study the impact of decoding strategies in section 3.1 and we believe that, being CM3leon our base model, their findings generalize to our model as well.  Regarding CLIP-reranking, we faced practical constraints in its application for interleaved generation. The CLIP text encoder has a limitation in the number of input tokens it can process, and our model often generates text captions preceding images that exceed this maximum token length. Therefore, while CLIP-reranking offers certain advantages, its applicability in our context is limited due to these inherent constraints.  We also wanted to keep the model functionality as homogeneous as possible across modalities, handling both text and image generation uniformly, since our primary aim here is not to push SoTA on image quality.  Clearly, more complex modality-specific methods could be employed to improve the benchmark numbers.
>
> [1] Scaling Autoregressive Multi-Modal Models: Pretraining and Instruction Tuning, Yu et al. 2023.
>
> > Can users control alpha_c, tao and other critical hyper-parameters during decoding? So the tool can be more balanced in generating text+image, or more leaning towards text or image generation.
>
> Yes, users have the flexibility to adjust critical hyperparameters such as alpha_c, tao, and others during the decoding process. This level of control allows for customization of the decoding process tailored to specific needs.
>
> We thank the reviewer for the suggestions that helped us improve the quality and clarity of the paper.

---

> ### Author Response · Authors · 2023-11-22
> **Follow up on response**
>
> Dear reviewer jvv1,
>
> We would like to bring to your attention that tomorrow (Nov 22) marks the conclusion of the rebuttal period. We are eager to know if our response has sufficiently addressed your concerns, or if there are additional questions you may have. We value the chance to engage in further discussion should there be any unresolved issues. In case our response has met your expectations, we kindly request that you consider modifying your initial score. Thank you once again for your attention and time.
>
> Warm Regards

---

### Official Review · Reviewer_qfqe · 2023-11-09

**Soundness:** 2 fair
**Presentation:** 2 fair
**Contribution:** 2 fair
**Rating:** 5
**Confidence:** 4

**Summary:**

JAM, novel methodologies for combining pretrained autoregressive image2text and text2image models, demonstrates remarkable new abilitys to generate interleaved image-text sequence.

**Strengths:**

This work proposes a novel approach to bridge two separate image2text and text2image models into a new unified model that can generate both image and text.

The resulting model demonstrates remarkable ability on generating interleaved image-text sequence under adequate qualitative evaluation compared with GILL.

**Weaknesses:**

**Method is restricted.**
1. The method is not general and nearly impossible for the community to follow. Because your method requires two identical image2text and text2image models, but nearly all available image-to-text and text-to-image models are of different architectures.
2. Thus, the only way for the community to test your method's effectiveness is to first pretraining two **identical** separate models. I think this is rather inefficient and prohibitive.

**Experiments seem too casual.**
1. Your paper focuses on multimodal models that generate both images and texts, but how can the performance of pure language tasks be your main table and experiments? **This table even becomes the most adequate part of your experiments**. This is not aligned with your motivation and focus.
2. Experiments about multimodal ability are too casual and not taken seriously. Your all multimodal experiments, except for the qualitative cases, are Table 2-5 on Page 7. Among them, three are ablation and Table 2 is your main table for multimodal performance. However, Table 2 uses MS-COCO as dataset and PPL as metric, which rarely serves as a **main** evaluation of multimodal ability. The baselines also lack a lot. The experiment quantity and quality is too bad to be a ICLR submission.

**Inadequate literature review.**

Your focus and motivation is to develop a multimodal model able to generate both image and text outputs. The field has witnessed an emergence of such kind of models. To name a few, Emu [1], SEED-LLaMA [2,3] and DreamLLM [3], etc. Among them, SEED-LLaMA [2,3] and DreamLLM [3] are recent work and can be arguably ignored, but Emu [1] is released even earlier than the blog of CM3Leon, the work you heavily follow. However, your literature review and discussion totally ignore such multimodal unified modeling work that generate both image and text, which I think are **the most closely related to your work** and should not be ignored.

[1] Generative pretraining in multimodality

[2] Planting a seed of vision in large language model

[3] Making LLaMA SEE and Draw with SEED Tokenizer

[3] Dreamllm: Synergistic multimodal comprehension and creation

**Questions:**

My rating is 4. But as only 3 and 5 are available, I choose 5 for the novelty in combining two separate models to empower better multimodal capability. But this work have too many deficiencies too.

---

> ### Author Response · Authors · 2023-11-17
> **Response to reviewer qfqe**
>
> We thank the reviewer for the insightful suggestions and valuable feedback, we address the different highlighted weaknesses separately:
>
> > Method is restricted.
> > 1. The method is not general and nearly impossible for the community to follow. Because your method requires two identical
> > image2text and text2image models, but nearly all available image-to-text and text-to-image models are of different architectures.
> > 2. Thus, the only way for the community to test your method's effectiveness is to first pretraining two identical separate models. I think
>     this is rather inefficient and prohibitive.
>
> We thank the reviewer for raising this concern.  Indeed we debated this point while designing our experiments, but decided to go with an identical architecture to be able to compare to baseline methods such as weight averaging and width concatenation.  However, our proposed cross-model fusion method does not rely on this limitation.  It can be applied to any two base models, as long as they both use a transformer architecture.  We acknowledge that we did not run further experiments to verify the method more generally, which is a limitation of this work due to resource and time constraints.  Being an early work in this area, we chose to focus on an experimental setup which allows us more control over variables, so as to better inform future architecture choices in this domain.  Having established the cross-model fusion as a viable technique, we hope to focus future work on scaling and demonstrating this architecture in more practical settings.
>
> > Experiments seem too casual.
> > 1. Your paper focuses on multimodal models that generate both images and texts, but how can the performance of pure language
> > tasks be your main table and experiments? This table even becomes the most adequate part of your experiments. This is not aligned
> > with your motivation and focus.
>
> We understand the reviewer's concern regarding our evaluation setting. The rationale behind this was to first ensure that after the first alignment phase our model maintains the foundational capabilities of the original models post-merging. Our intention was to establish that the joint training does not detrimentally affect the individual performance of each component, a necessary precondition for effective multimodal integration.  While Table 1 and Table 2 are not our main results, we find that they are useful in discriminating between fusion approaches, since most baseline approaches will actually cause degradation in single-modality benchmarks.
>
> > 2. Experiments about multimodal ability are too casual and not taken seriously. Your all multimodal experiments, except for the
> > qualitative cases, are Table 2-5 on Page 7. Among them, three are ablation and Table 2 is your main table for multimodal performance.
> > However, Table 2 uses MS-COCO as dataset and PPL as metric, which rarely serves as a main evaluation of multimodal ability. The
> > baselines also lack a lot. The experiment quantity and quality is too bad to be a ICLR submission.
>
> We acknowledge the complexity involved in evaluating text-to-image models, which guided our decision to utilize the MS-COCO dataset and Perplexity (PPL) as our primary metrics. We believe that this metric is a strong indicator of downstream performance for common downstream tasks of multimodal vision language models such as text-to-image generation, image captioning and so on, as evidenced by prior work.
> Our goal with this evaluation was to facilitate comparisons between our model and other established autoregressive multimodal architectures, including RA-CM3, CM3, and CM3leon. This was done to ensure that the integration of our two backbones does not compromise their individual performance.
> Unlike recent works such as Emu [1], which evaluates either text-to-image or image-to-text generation individually due to their model's constraint of producing only one modality at a time.  Our model uniquely generates multiple modalities simultaneously. Our focus is not on surpassing the text-to-image generation metrics such as FID, instead, our evaluation aims to demonstrate that our model maintains a comparable image-text performance level to our foundational model, CM3leon. However, we recognize that evaluating our model's unique capability of interleaved generation is challenging, as there are no established metrics for this task.  To address the absence of established quantitative metrics for interleaved simultaneous generation, we have placed a significant emphasis on thorough qualitative analysis.  The primary aim of our paper is not necessarily to achieve state-of-the-art results on existing single modality generation metrics,  but to demonstrate the successful merging of two distinct large-scale trained autoregressive models. This integration not only preserves the performance in their respective domains but also unlocks new capabilities in the realm of interleaved generation.

---

> > ### Author Response · Authors · 2023-11-17
> > **Response to reviewer qfqe (pt.2)**
> >
> > > Inadequate literature review.
> > > Your focus and motivation is to develop a multimodal model able to generate both image and text outputs. The field has
> > > witnessed an emergence of such kind of models. To name a few, Emu [1], SEED-LLaMA [2,3] and DreamLLM [3], etc. Among
> > > them, SEED-LLaMA [2,3] and DreamLLM [3] are recent work and can be arguably ignored, but Emu [1] is released even earlier
> > > than the blog of CM3Leon, the work you heavily follow. However, your literature review and discussion totally ignore such
> > > multimodal unified modeling work that generate both image and text, which I think are the most closely related to your work
> > > and should not be ignored.
> >
> >
> > We thank the reviewer for pointing out the omission of key works such as Emu [1]. We have revised our literature review to include a detailed comparison with Emu, recognizing its pioneering role in multimodal generation. This addition will provide a clearer understanding of how our work contributes uniquely to the field of multimodal models. While we omit SEED-LLaMA and DreamLLM since they were released close to our submission date and they can be regarded as concurrent submission. Nonetheless we think our approach is consistently different from [2] [3] [4], and it brings its unique contribution to the field of multimodal modeling, especially given our focus to interleaved generation. Moreover, these works rely on a pretrained Stable Diffusion model for image generation, similarly to what has been presented in GILL [5], while our framework is completely autoregressive.
> >
> >
> > [1] Generative pretraining in multimodality.
> > [2] Planting a seed of vision in large language model.
> > [3] Making LLaMA SEE and Draw with SEED Tokenizer.
> > [4] Dreamllm: Synergistic multimodal comprehension and creation.
> > [5] Generating Images with Multimodal Language Models.
> >
> >
> > We thank the reviewer for the suggestions that helped us improve the quality and clarity of the paper.

---

> ### Author Response · Authors · 2023-11-22
> **Follow-up on response**
>
> Dear reviewer qfqe,
>
> We would like to bring to your attention that tomorrow (Nov 22) marks the conclusion of the rebuttal period. We are eager to know if our response has sufficiently addressed your concerns, or if there are additional questions you may have. We value the chance to engage in further discussion should there be any unresolved issues. In case our response has met your expectations, we kindly request that you consider increasing your initial score. Thank you once again for your attention and time.
>
> Warm Regards

---

### Meta-Review · Area_Chair_ZY8Y · 2023-12-06

**Metareview:**

2x BA and 2x BR. This paper proposes to bridge pre-trained autoregressive text decoder and text-to-image decoder into one for multimodal generation. The reviewers consistently appreciate the (1) important topic, (2) novel approach, and (3) SOTA results. Their minor concerns about the (1) unclear presentation and (2) insufficient technical details have been amended by the rebuttal and pdf revisions. The AC leans to accept this submission.

**Justification For Why Not Higher Score:**

N/A

**Justification For Why Not Lower Score:**

N/A

---

### Decision · Program_Chairs · 2024-01-16

Accept (poster)